# Amortized Normalizing Flows for Transcranial Ultrasound with Uncertainty Quantification

**Rafael Orozco**[1]                                    rorozco@gatech.edu
**Mathias Louboutin**[1]                           mlouboutin3@gatech.edu
**Ali Siahkoohi**[2]                                          alisk@rice.edu
**Gabrio Rizzuti**[3]                               g.rizzuti@umcutretcht.nl
**Tristan van Leeuwen**[4]                          T.van.Leeuwen@cwi.nl
**Felix Herrmann**[1]                           felix.herrmann@gatech.edu

[1]*Georgia Institute of Technology* [2]*Rice University* [3]*University Medical Center Utrecht*
[4]*Centrum Wiskunde & Informatica*

**Editors:** Accepted for publication at MIDL 2023

## Abstract

We present a novel approach to transcranial ultrasound computed tomography that utilizes normalizing flows to improve the speed of imaging and provide Bayesian uncertainty quantification. Our method combines physics-informed methods and data-driven methods to accelerate the reconstruction of the final image. We make use of a physics-informed summary statistic to incorporate the known ultrasound physics with the goal of compressing large incoming observations. This compression enables efficient training of the normalizing flow and standardizes the size of the data regardless of imaging configurations. The combinations of these methods results in fast uncertainty-aware image reconstruction that generalizes to a variety of transducer configurations. We evaluate our approach with in silico experiments and demonstrate that it can significantly improve the imaging speed while quantifying uncertainty. We validate the quality of our image reconstructions by comparing against the traditional physics-only method and also verify that our provided uncertainty is calibrated with the error.

**Keywords:** Invertible Networks, Medical Imaging, Bayesian Estimation, Uncertainty Quantification, Physics and Machine Learning Hybrid

## 1. Introduction

Transcranial ultrasound computed tomography (TUCT) is a non-invasive, non-toxic imaging technique that aims to create images of internal brain tissue by transmission and reception of acoustic waves (Dines et al., 1981). Its clinical applications range from hemorrhage detection to tumour imaging Becker et al. (1994). Previous approaches to TUCT utilized time-of-flight methods such as B-mode ultrasound (Smith et al., 1978). These methods are limited in their imaging resolution for a variety of reasons, the foremost of which is due to their approximate treatment of wave physics (Williamson, 1991). Following the pioneering work by Guasch et al. (2020), it was shown that modeling all aspects of the acoustic wavefield enables high-resolution imaging of brain structures and anomalies. Since then, many works Taskin et al. (2020); Marty et al. (2021); Tong et al. (2022); Cudeiro-Blanco et al. (2022); Bates et al. (2022) are demonstrating increasing evidence from both in silico

and controlled laboratory experiments that these full wavefield methods are capable of producing reliable brain images bringing this novel approach closer to clinical viability. These full wavefield methods are denoted full-waveform inversion (FWI) and are adapted from sophisticated seismic imaging methods (Virieux and Operto, 2009; Tarantola and Valette, 1982). On the downside, FWI methods are computationally intensive since they require the application of forward and gradient operators related to expensive partial differential equation (PDE) solutions. This limits the clinical use of FWI methods towards TUCT since they can take up 36 hours to form an image Guasch et al. (2020). In addition, the imaging process is affected by incomplete measurements, noise and other sources of uncertainty that can limit the accuracy and reliability of TUCT. To alleviate these problems and facilitate the adoption of this new imaging modality, we propose a data-driven approach to TUCT that leverages normalizing flows to dramatically improve the speed of imaging and provide uncertainty quantification (UQ). While deep learning has tremendous potential in accelerating computational imaging (Ongie et al., 2020), we identify the limitation that ultrasound measurements in TUCT are impractically large and contain complex relationships that are difficult to undo without the aid of the underlying physics model. We propose to solve these problems by using a physics-informed summary function that takes the physical wave model into account. For our data recording setup, this summary compresses the size of observations by a factor of $70\times$ allowing the use of GPU hardware accelerators. Figure 1 contains a schematic of our full proposed framework.

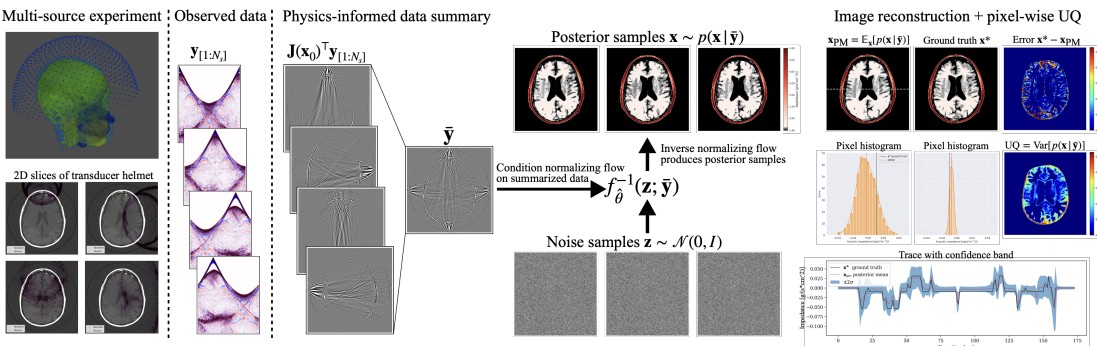

Figure 1: Proposed transcranial image reconstruction framework with normalizing flows for uncertainty quantification.

## 2. Methods

### 2.1. Ultrasound modeling

Our imaging approach, solves the inverse problem of finding acoustic properties of internal brain tissue that match observed ultrasound data. To model the propagation of ultrasound waves through a human skull, we use the scalar acoustic wave equation with variable density.

We express the data recording process (solving the wave equation in Equation (4) of the Appendix, followed by a restriction of the wavefield to the transducer locations) by the

discrete nonlinear operator, $\mathcal{F}$, acting on the $i$th known source represented by the vector $\mathbf{q}_i$. This nonlinear forward model is parameterized by the unknown acoustic impedance, discretized on a $N_x \times N_y$ grid (with $N_x = 512, N_y = 512$) and represented by the vector, $\mathbf{x} \in \mathbb{R}^{N_x \times N_y}$. For each source, $\mathbf{q}_i$, data is collected at $N_r$ receivers for $N_t$ time steps yielding

$$\mathbf{y}_i = \mathcal{F}(\mathbf{x})\mathbf{q}_i + \boldsymbol{\varepsilon}_i, \tag{1}$$

with $\mathbf{y}_{[1:N_s]} = \{\mathbf{y}_i\}_{i=1}^{N_s}$ being the full observation over $N_s$ sources. To account for errors in the measurements, an additive noise term is included as $\boldsymbol{\varepsilon} \in \mathbb{R}^{N_r \times N_t}$. Typical 3D hardware setups have $N_s = 1024$ sources and for our 2D simulation we use up to $N_s = 32$ sources. This makes the full observation $\mathbf{y}_{[1:N_s]} \in \mathbb{R}^{N_r \times N_t \times N_s}$. Figure $2(b)$ shows data of a single source experiment with the acoustic impedance shown in Figure $2(a)$. In our setup, we model $N_r = 256$ transducer receivers around the skull, of which $N_s = 32$ also act as sources. They record for $N_t = 2377$ time steps. Given observed transcranial ultrasound data, $\mathbf{y}_{[1:N_s]}$, our aim is to invert for internal structures $\mathbf{x}$. We solve this inverse problem in a Bayesian framework so uncertainty due to incomplete measurements, modeling errors, and noise, can be quantified systematically.

## 2.2. Bayesian transcranial ultrasound

Upon receiving observations $\mathbf{y}$, solving a Bayesian inverse problem involves sampling the conditional distribution of $\mathbf{x}$ given $\mathbf{y}$ (Tarantola, 2005). This conditional distribution $p(\mathbf{x}|\mathbf{y})$ is called the posterior distribution. This posterior gives the full set of acoustic models $\mathbf{x}$ that explain the observations $\mathbf{y}$. To form an image reconstruction, one can use posterior samples to calculate high-quality point estimates such as the maximum a posteriori (MAP) and the minimum mean squared error (MMSE) estimator, while also providing uncertainty of those estimates. In general, the posterior distribution $p(\mathbf{x}|\mathbf{y})$ is computationally costly to sample from. Traditional methods like Markov chain Monte Carlo (McMC) require thousands of iterations, each of which needs to evaluate the expensive forward operator $\mathcal{F}$ (Martin et al., 2012; Curtis and Lomax, 2001). This makes these methods impractical for clinical use scenarios that require fast results (Bauer et al., 2013). In this paper, we suggest a variational inference method (Jordan et al., 1999) that accomplishes fast posterior sampling by exploiting the distribution learning capabilities of generative models (Ruthotto and Haber, 2021). We will explain how our method derives from amortized density estimation where an expensive offline pre-training phase leads to fast posterior sampling at inference time for any in-distribution observation.

## 2.3. Amortized normalizing flows for posterior distribution sampling

Our goal is to sample from the distribution $p(\mathbf{x} \mid \mathbf{y})$ so that we can study the variation of different $\mathbf{x}$ that explain the observed data $\mathbf{y}$. Normalizing flows are a deep learning technique that have shown to be capable of learning to sample from complicated distributions (Dinh et al., 2014, 2016). This method works by learning to map samples from the target distribution to standard white Gaussian noise using an invertible neural network $f_\theta$ with learned layers parameterized by $\theta$. Once trained, the inverse of the network $f_{\hat{\theta}}^{-1}$ is evaluated on realizations of standard white Gaussian noise to generate new samples from the target distribution. Due to multi-scale transformations, normalizing flows scale favorably with

dimension of the target distribution and allow for fast sampling (Bond-Taylor et al., 2021) making them a good candidate for our high-dimensional medical image reconstruction task.

The posterior distribution $p(\mathbf{x} \mid \mathbf{y})$ we want to sample from is a conditional distribution so we use conditional normalizing flows (Ardizzone et al., 2019; Winkler et al., 2019). These learn to sample from a distribution conditioned on an observation $\mathbf{y}$ by minimizing the following objective:

$$\hat{\theta} = \arg\min_{\theta} \frac{1}{N} \sum_{n=1}^{N} \left( \|f_\theta(\mathbf{x}^{(n)}; \mathbf{y}^{(n)})\|_2^2 - \log |\det \mathbf{J}_{f_\theta}| \right) \tag{2}$$

where $\mathbf{J}_{f_\theta}$ is the Jacobian of the network and $\{(\mathbf{x}^{(n)}, \mathbf{y}^{(n)})\}_{n=1}^{N}$ are training pairs given by Equation (1) and samples $\mathbf{x} \sim p(\mathbf{x})$ drawn from the prior. Intuitively, Equation (2) learns the posterior distribution by maximizing the likelihood of the $\mathbf{x}$ conditioned on $\mathbf{y}$ under the normalizing transformation $f_\theta$. The first term is the likelihood in Normal distribution ($\ell_2$ norm). Because the transformation is invertible, the change of variables formula is used to evaluate the likelihood in Normal space by controlling for volume changes caused under the normalizing transformation $f_\theta$ as quantified by the second Jacobian term. Mathematically, Equation (2) minimizes the Kullback-Leibler divergence between the learned posterior and the true posterior (Radev et al., 2020; Kovachki et al., 2020; Siahkoohi et al., 2022). Crucially to our application, this method learns the posterior in an amortized fashion since it minimizes the objective over a distribution of $\mathbf{y}$. After training, the conditional normalizing flow can sample the posterior for unseen $\mathbf{y}$ at the cheap cost of passing noise through the inverse network. See Figure 1 for a schematic of the sampling process from noise.

Normalizing flows, due to their architecture, have closed-form inverses (up to numerical precision), that cost the same as forward evaluation and the term $|\det \mathbf{J}_{f_\theta}|$ can be efficiently calculated. In general, training pairs needed to optimize Equation (2) are generated in the simulation-based inference framework (Cranmer et al., 2020) but for our ultrasound application, $\mathbf{y}$ is complicated acoustic data and is too large for GPU training thus we explore a physics-informed method to extract important features and compress its size.

## 2.4. Physics-informed summary statistic

For our ultrasound application, we identify three difficulties of working with acoustic data $\mathbf{y}$. First, the observation for all sources $\mathbf{y}_{[1:N_s]}$ is too large ($N_t \times N_r \times N_s \approx 19 \times 10^6$) to fit in a GPU for training. Second, different experimental configurations (i.e. varying number of sources) change the size of observations meaning generalization on data space requires sophisticated architectures (Radev et al., 2020). Finally, imaging complicated structures directly from acoustic data is a difficult task (Orozco et al., 2022). These considerations motivate the need of a function $h$ that reduces the size and "summarizes" the observation $\bar{\mathbf{y}} = h(\mathbf{y}_{[1:N_s]})$ while preserving information it carries about $\mathbf{x}$. These summaries are formally known as summary statistics (Deans, 2002; Radev et al., 2020). In the context of maximum likelihood estimation, Alsing and Wandelt (2018) proposed the score of the likelihood as a summary statistic. This score is defined as the gradient of the log-likelihood $\mathcal{L} = \log p(\mathbf{y} \mid \mathbf{x})$ with respect to $\mathbf{x}$. Alsing and Wandelt (2018) proved that the score is asymptotically maximally informative of $\mathbf{x}$. Inspired by this approach, we explore using the score as a

summary function for posterior sampling. We assume a Gaussian noise model leading to the gradient being the Jacobian adjoint $\mathbf{J}^\top$ on the data residual:

$$\bar{\mathbf{y}} = h(\mathbf{y}_{[1:N_s]}) := \nabla_{\mathbf{x}_0}\mathcal{L} = \sum_{i=1}^{N_s} \mathbf{J}(\mathbf{x}_0, \mathbf{q}_i)^\top (\mathcal{F}(\mathbf{x}_0)\mathbf{q}_i - \mathbf{y}_i) \tag{3}$$

where $\mathbf{x}_0$ is a starting point at which the gradient is calculated. Note, Equation (3) involves evaluating the forward physical model $\mathcal{F}$ and its Jacobian adjoint $\mathbf{J}^\top$. Thus this summary is informed by the physics (domain knowledge). As a result, the summarized data $\bar{\mathbf{y}}$ lives in the reduced $N_x \times N_y$ image space (reduction factor of about 70). According to Fluri et al. (2021), the informativeness of this summary statistic also implies that $p(\mathbf{x} \mid \mathbf{y}) = p(\mathbf{x} \mid \bar{\mathbf{y}})$ thus we propose to use the same conditional distribution learning objective as Equation (2) but replace the data $\mathbf{y}$ with the summary $\bar{\mathbf{y}}$. See Algorithm 1 in the Appendix for our full training process. The technical assumptions for the informativeness of this summary statistic are discussed in Appendix 4.5 alongside studies to understand the effect of deviations from the assumptions. One of the assumptions is that the starting point $\mathbf{x}_0$ needs to be carefully chosen as it will affect how informative the summary statistic will be. For our application, $\mathbf{x}_0$ is the acoustically correct model of the skull bone and a constant acoustic model inside the skull since the soft tissues inside the skull are the clinically relevant structures we care to image. Inclusion of the skull is needed so that the physical operators create meaningful results that inform the posterior. In practice, acoustic values of skull bone can be calculated from CT scans (Aubry et al., 2003). See Figure 2(c) for an example of $\mathbf{x}_0$ and Figure 2(d) for the physics-informed summary $\bar{\mathbf{y}}$ it creates.

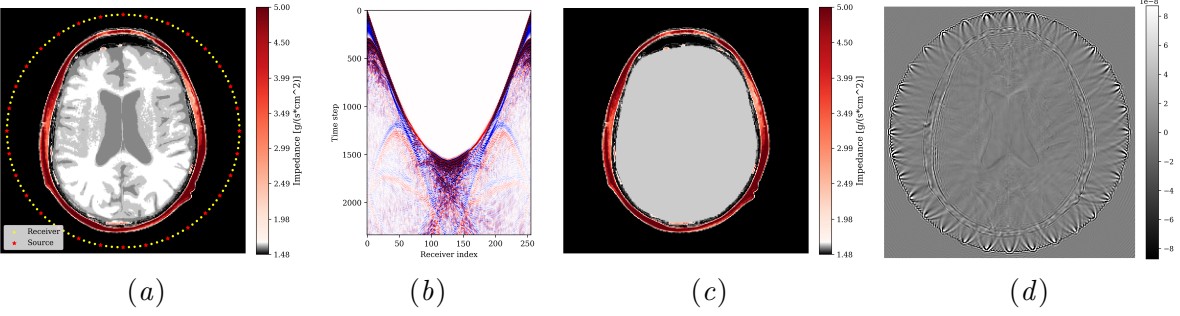

$(a)$        $(b)$        $(c)$        $(d)$

Figure 2: 2D transcranial ultrasound imaging setup. (a) Ground truth acoustic impedance $\mathbf{x}^*$ including source/receiver layout; (b) Observed data $\mathbf{y}_i$ from a single source; (c) Starting model, $\mathbf{x}_0$, which includes detailed acoustic information on the skull; (d) summarized data $\bar{\mathbf{y}}$ for all $N_s=32$ sources.

While previous work has used the adjoint operator and pseudo-inverse to summarize data (Adler and Öktem, 2018; Adler et al., 2022) to the best of our knowledge this is the first work that explores based on theoretical arguments the use of the score of the likelihood as a summary statistic for direct posterior sampling in a inverse problem with an expensive physics-based nonlinear operator.

## 3. Experiments and Results

### 3.1. Normalizing flow training

To create training pairs, we require samples from the prior distribution $p(\mathbf{x})$ of ground truth brain acoustic impedance models. In Appendix 4.2, we detail our automatic process for deriving acoustic models from the FastMRI dataset (Zbontar et al., 2018). For training and testing, we use 250 3D acoustic brains models each containing 11 $512 \times 512$ slices. Out of these, we used 90% for training, 5% for validation and 5% for testing. We simulated the forward wave propagation $\mathcal{F}$ from Equation (4) and its Jacobian adjoint $\mathbf{J}^\top$ using Devito (Luporini et al., 2020; Louboutin et al., 2019) and JUDI (Witte et al., 2019).

The conditional normalizing flow is implemented with InvertibleNetworks.jl (Witte et al., 2020). Each epoch takes about 20 minutes and we trained for a total of 18 hours on a 32GB A100 GPU. We did not observe over fitting on the validation set (Appendix Figure 6).

### 3.2. Image reconstruction from posterior samples

Once trained, our conditional normalizing flow can generate samples from the posterior with Algorithm 2. The computational cost of posterior sampling is dominated by the calculation of the physics-informed summary $\bar{\mathbf{y}}$. This takes $\approx 1$ second per source and 44.8 seconds in total for all 32 sources (on 4 core Intel Skylake CPU). This calculation only needs to be done once per ultrasound experiment after which many posterior samples can be generated each at the cheap cost of one inverse network evaluation (20ms/sample). With these posterior samples, statistical point estimates can be calculated including the minimum mean squared error (MMSE) estimator given by the posterior/conditional mean $\mathbf{x}_{\mathrm{PM}} = \mathbb{E}_\mathbf{x}[\,p(\mathbf{x} \mid \bar{\mathbf{y}})]$ that serves as our image reconstruction. For UQ, we look at the intra-sample variation between posterior samples. To visualize UQ on the entire image reconstruction we use the posterior variance $\mathrm{Var}[\,p(\mathbf{x} \mid \bar{\mathbf{y}})]$. The posterior mean (and variance) is calculated by approximating their expectations with an average over $N_{\mathrm{post}} = 128$ posterior samples

$$\mathbf{x}_{\mathrm{PM}} = \mathbb{E}_\mathbf{x}[\,p(\mathbf{x} \mid \bar{\mathbf{y}})] \approx \frac{1}{N_{\mathrm{post}}} \sum_{i=1}^{N_{\mathrm{post}}} \mathbf{x}_i \ \text{ where } \mathbf{x}_i = f_{\hat{\theta}}^{-1}(\mathbf{z}_i; \bar{\mathbf{y}}) \text{ and } \mathbf{z}_i \sim \mathcal{N}(0,\, I).$$

See Appendix 4.7 for an analysis of the quality of $\mathbf{x}_{\mathrm{PM}}$ as the number of posterior samples increases. In this work, we concentrate on the posterior mean because it is the estimator with minimal mean squared error (Whang et al., 2021). Figure 3 contains an example of the input and output of the proposed image reconstruction algorithm including UQ.

To assess the performance of our reconstruction, $\mathbf{x}_{\mathrm{PM}}$, we compare with two baseline methods, namely physics-only FWI, yielding $\mathbf{x}_{\mathrm{FWI}}$ obtained by gradient descent, and a supervised U-Net $\mathbf{x}_{\mathrm{UNET}}$ (Ronneberger et al., 2015) trained on the same $N$ data pairs $\{(\mathbf{x}^{(n)}, \bar{\mathbf{y}}^{(n)})\}_{n=1}^N$ as our method. Compared to the learned methods, which incur off-line training costs prior to inference, FWI is computationally intensive since it requires $\sim 40$ calls to the forward and gradient for each source while our method only requires one gradient per source. Refer to Appendix 4.3 for FWI and network training hyperparameters.

From Figure 4, we make the following observations: (i) our result contains fewer artifacts compared to FWI; (ii) it performs better than U-Net; (iii) it captures the full posterior

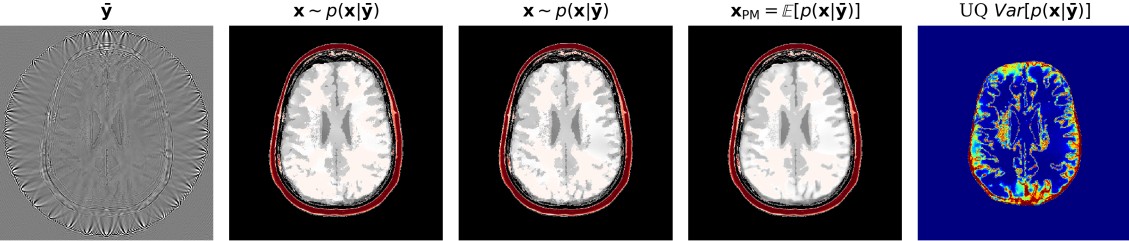

Figure 3: Image reconstruction with UQ using our method including samples from the posterior.

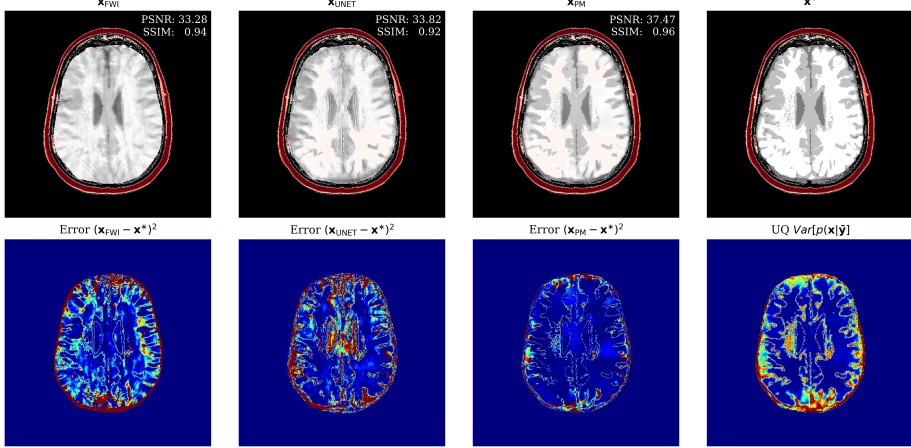

Figure 4: Comparison with physics-only and data-only methods of FWI and supervised U-Net. Note that areas in our pointwise variance correlate well with areas of high error.

| Method | Timing (seconds) | PSNR ↑ | SSIM ↑ | RMSE ↓ |
|---|---|---|---|---|
| FWI ($\mathbf{x}_{\text{FWI}}$) | 2100 | 33.25 | 0.9450 | 0.0215 |
| Supervised UNet ($\mathbf{x}_{\text{UNET}}$) | **44.8 + 0.02** | 35.63 | 0.9332 | 0.0168 |
| Our posterior mean ($\mathbf{x}_{\text{PM}}$) | 44.8 + 3.23 | **38.67** | **0.9646** | **0.0119** |

Table 1: Image reconstruction timing and quality metric comparison

yielding pointwise variances that correlate well with error; (iv) due to averaging over posterior samples our result blurs a few details as compared to FWI. For a more quantitative comparison of the reconstruction quality, refer to Table 1 in which the average quality metrics for peak signal to noise ratio (PSNR); structural similarity index metric (SSIM); and root mean squared error (RMSE) are computed from 50 unseen test slices. Our method shows high performance on all metrics while keeping the online inference time significantly lower than the FWI method. For more direct comparison, we avoided measurement noise.

### 3.3. Generalization over experimental configurations

In Figure 5, we show how our method generalizes over different source configurations. Aside from handling different acquisition constraints, practitioners can also quickly prototype different configurations to decide which one meets their threshold of uncertainty.

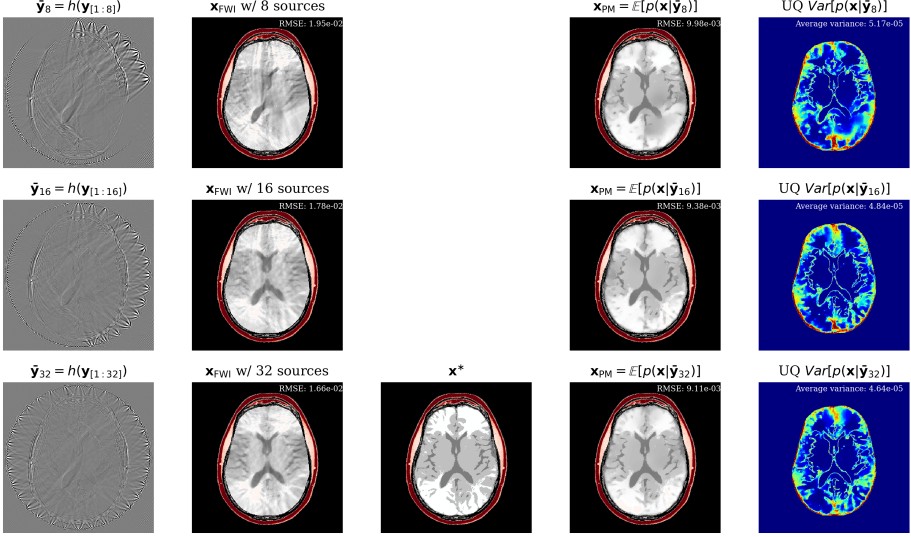

Figure 5: Generalization over different imaging configurations. The three FWI results took ≈1.5 compute hours but the three posterior means and UQ were calculated in ≈3 minutes. We observe that our method shows better results than the pure-physics FWI when there is less source coverage.

**Related work:** The gradient we calculate for our summary statistic is connected to reverse time migration from seismic imaging (Baysal et al., 1983). For accessing uncertainty information in TUCT, (Bates et al., 2022) use the mean-field Gaussian approximation. Their method uses gradient descent with many expensive forward/gradient calls and assumes a Gaussian prior on the ground truth images while neglecting correlations between pixels. Our work, instead makes no underlining assumptions on the posterior/prior distributions and requires only one set of forward/gradient calls during inference. (Radev et al., 2020) explored learned summary statistics for posterior inference. Here we exploit knowledge of the underlying physics by introducing physics-informed summary statistics. Instead of including physics in learned simulations as in physics-informed neural networks, we include the physics in the data summary, which makes sense when dealing with inverse problems where observed data serves as input.

**Future work:** Normalizing flows are likelihood models so they allow for natural anomaly detection (Gudovskiy et al., 2022). We will explore the possibility of evaluating our method on brains with anomalies for automatic detection of tumors or hemorrhages.

We highlight that our method assumes access to good starting points $\mathbf{x}_0$. In Appendix 4.5 we show that as this starting point gets worse then the physics-informed summary statistic fails to inform the posterior. This results in degradation of the samples. We see this result as a quality assurance since our generative model does not falsely generate realistic but wrong samples. This is a limitation of gradient approaches in nonlinear problems as demonstrated by FWI also failing for the poor starting points Appendix Figure 9. In future works, we would like to find ways to be robust against poor starting points.

**Conclusions:** The application of machine-learning methods and systematic uncertainty quantification to ultrasound imaging has been extremely challenging because of the high-dimensionality and high computational costs associated with handling the correct wave physics. Through the combination of conditional normalizing flows with physics-informed summary statistics, we arrive at a formulation capable of producing high-fidelity images with uncertainty quantification. By incurring an off-line pretraining cost, our method is faster than traditional physics-only methods.

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

## 4. Appendix

### 4.1. Wave equation modeling

The wave equation we use is

$$\frac{1}{\rho(x,y)c(x,y)^2}\frac{\partial^2}{\partial t^2}u(x,y,t) - \nabla \cdot \frac{1}{\rho(x,y)}\nabla u(x,y,t) = q(t,x,y). \tag{4}$$

In Equation (4): $\rho(x,y)$ represents density as a function of space, $c(x,y)$ is acoustic velocity, $u(x,y,t)$ is the acoustic pressure as a function in space and time, $\nabla$ is the derivative in space and $q(t,x,y)$ is the acoustic source that is defined by the experimental transducer. For our experiments, the transducers where impulsed using a 3-cycle burst with central frequency of 400kHz. All values are defined on a discrete grid with spacing of 0.5[mm].

### 4.2. Generating acoustic models from FastMRI

To make acoustic training models, we start from the FastMRI dataset (Zbontar et al., 2018) that contains MRI images of human brains. There is no immediate relationship between MRI intensity and acoustic values. As a heuristic, we took the acoustic values of the main brain tissues (Brain Grey Matter = $1505[m/s]$, Cerebellum White

Matter=1552[$m/s$], Blood Veins=1578[$m/s$]) and (Brain Grey Matter=1044.5[$Kg/m^3$], Cerebellum White Matter = 1041.5[$Kg/m^3$], Blood Veins=1049.8[$Kg/m^3$]), and used k-means to assign acoustic values to MRI intensities. This process is automatic and we generated 2D slices of acoustic values from 250 brains. In future work, we will explore more complicated workflows to produce acoustic models required for training.

## 4.3. Training details and FWI setup

We trained the conditional normalizing flow using ADAM optimizer (Kingma and Ba, 2014) with learning rate of 0.001. We did not find the need to taper the learning rate. The mini-batch size was 8. The supervised UNet uses the same implementation as Ronneberger et al. (2015) with 5 downsampling levels. The UNet was trained with ADAM and a learning rate of 0.0001 and needed exponential decay for stable training. We implemented FWI on acoustic velocity by performing stochastic gradient descent on the $\ell_2$ misfit until convergence or 35 minutes had elapsed. To accelerate convergence, we used a backtracking line-search and box bounds projection onto the minimum and maximum acoustic values of water and bone.

---

**Algorithm 1** Pre-training phase

---

Need: N samples from prior $p(\mathbf{x})$
**for** $i \in 1 : N$ **do**
    Sample from prior: $\mathbf{x} \sim p(\mathbf{x})$; Sample from noise model: $\epsilon \sim p(\epsilon)$
    Generate synthetic observation by solving forward PDE Equation (4): $\mathbf{y}_i = \mathcal{F}(\mathbf{x})\mathbf{q}_i + \epsilon$
    Generate starting point: $\mathbf{x}_0 = \mathbf{extractskull}(\mathbf{x})$
    Summarize data with physics-informed gradient: $\bar{\mathbf{y}} = \sum_{i=1}^{N_s} \mathbf{J}(\mathbf{x}_0, \mathbf{q}_i)^\top (\mathcal{F}(\mathbf{x}_0)\mathbf{q}_i - \mathbf{y}_i)$
    Add pairs to dataset: $\mathcal{D}_i = (\mathbf{x}, \bar{\mathbf{y}})$
**end**
**while** *normalizing flow $f_\theta$ is not converged* **do**
    Evaluate $f_\theta$ on dataset $\mathcal{D}$ using Equation (2) and update $\theta$ using backpropagation
**end**

---

---

**Algorithm 2** Amortized posterior inference (given unseen observation $\mathbf{y}_{[1:N_s]}$)

---

Need: starting point $\mathbf{x}_0$
Calculate gradient summary $\bar{\mathbf{y}} = \sum_{i=1}^{N_s} \mathbf{J}(\mathbf{x}_0, \mathbf{q}_i)^\top (\mathcal{F}(\mathbf{x}_0)\mathbf{q}_i - \mathbf{y}_i)$
Sample $N_{post}$ Gaussian normal noise $\mathbf{z} \sim \mathcal{N}(0, I)$
Pass $\mathbf{z}$'s through inverse of normalizing flow $f_{\hat{\theta}}^{-1}(\mathbf{z}; \bar{\mathbf{y}})$ to generate posterior samples.

---

## 4.4. Evaluating sensitivity to size of training dataset

Since our method is Bayesian, its UQ results depend on how well it has learned the prior from training examples. In the case of conditional normalizing flows the prior is not explicitly accessible from the network since the network directly learns to sample the conditional distribution. Nonetheless, we would like to gain intuition on the effect of more training samples on the methods performance. In Figure 7, we demonstrate the effect of increasing

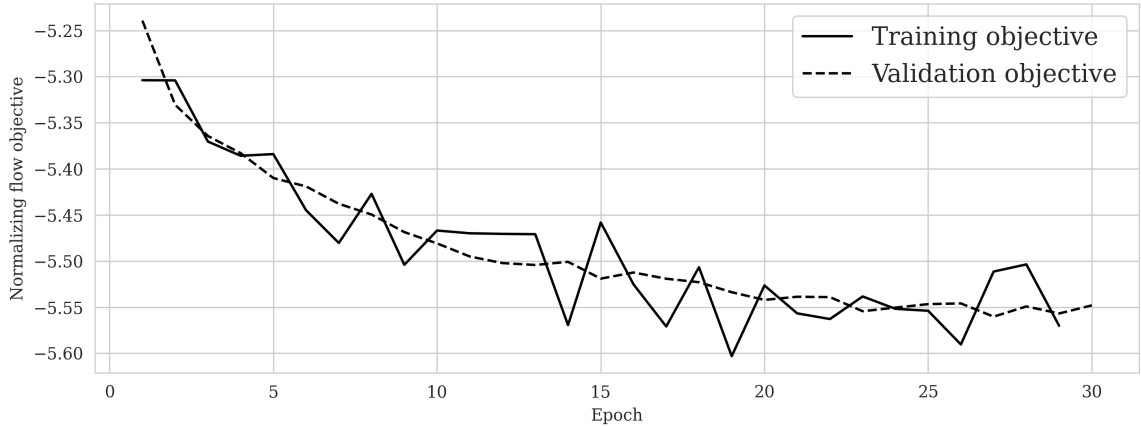

Figure 6: Training log of normalizing flow with values from a leave-out validation set for early stopping.

the training dataset size, on the posterior mean quality and on the UQ map that is produced. We observe from Figure 7 that as training samples increase, the posterior mean gets closer to the ground truth and that the UQ map becomes more contracted. These observations are similar to what happens when we increase the amount of observed data as explained in Section 4.6.

### 4.5. Considerations on quality of physics-informed summary statistic

Alsing and Wandelt (2018) proved that the score is asymptotically maximally informative of $\mathbf{x}$. Informativeness is defined by the Fisher information that $\bar{\mathbf{y}}$ carries about $\mathbf{x}$. The term "asymptotically" refers to two conditions, firstly how close the assumed likelihood is to the ground truth one and secondly how close the starting point $\mathbf{x}_0$ is to the ground truth $\mathbf{x}$. Deviations from these two assumptions, will produce a summary statistic that is uninformative.

**Assumption 1. Assumed likelihood must be close to true likelihood:** The ground truth likelihood in our synthetic case is related to the noise model that we used to simulate our forward data. We used colored noise that was made by band-limiting Gaussian noise with the frequency content of the transducer wavelet. This would correspond to a noise model of non-isotropic Gaussian with covariance related to the noise level and the particular wavelet used. The likelihood we assumed to calculate the score is of an isotropic Gaussian with $\sigma = 1$. This is already a deviation from the true likelihood but we did not notice a degradation in quality.

**Assumption 2. Starting point $\mathbf{x}_0$ must be close to x:** An important consideration of our method is that since it is gradient based, we need a starting point at which to calculate the gradient. we assume that we have access to an acoustically correct model of the skull. Practically, we envisage that by using x-ray based computed tomography (CT) we will build an acoustic model of the patients skull that we can then use to invert for the acoustic

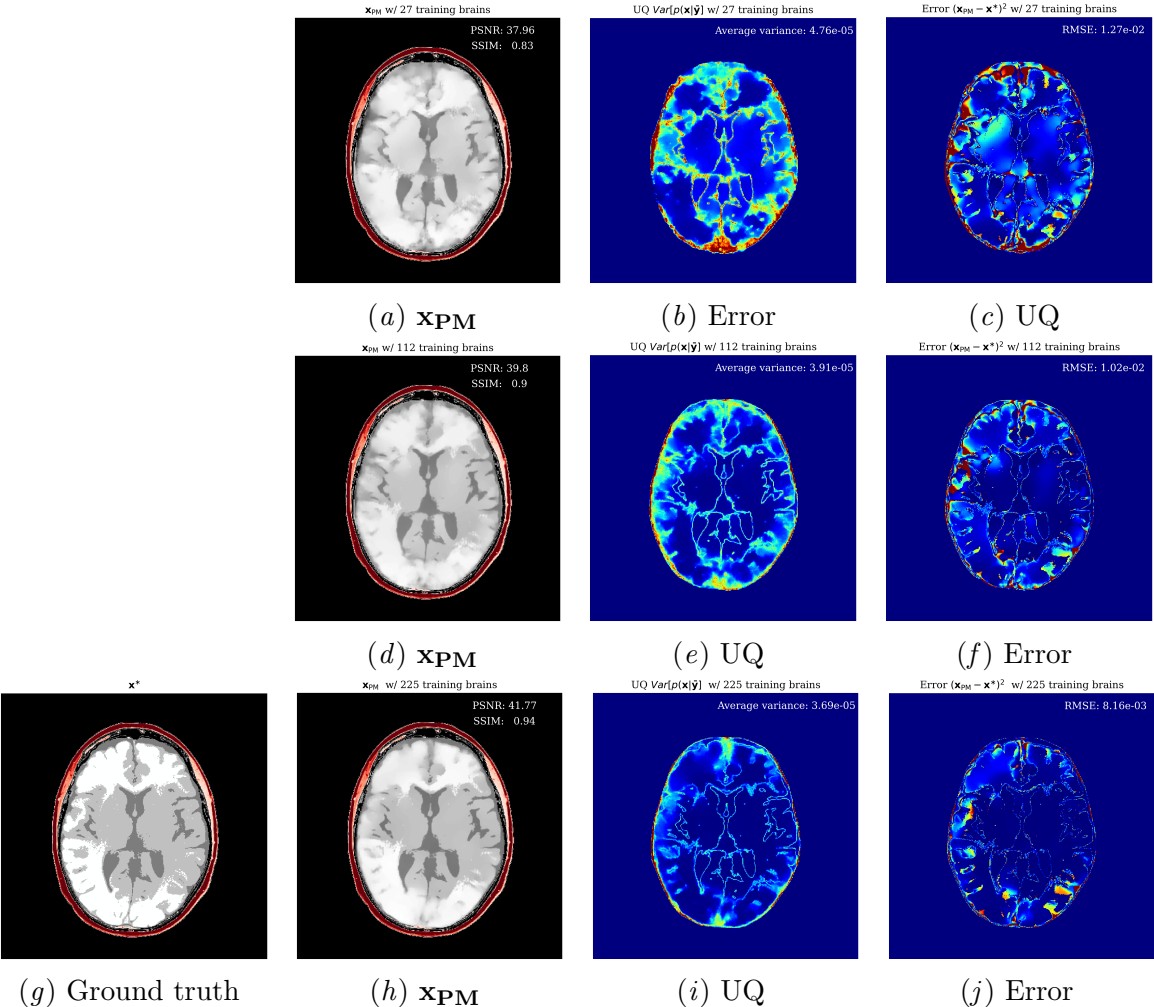

Figure 7: Increasing quantity of training samples increases the quality of the posterior mean and also decreases the average uncertainty.

properties of the inside brain tissue. The process of recovering acoustic properties of bone from CT measurements is well-documented (Aubry et al., 2003). Acoustic properties of bone are well-recovered by CT but the acoustic properties of soft tissue (the imaging goal of our method) are not.

Since our wave physics model is nonlinear, the process will be particularly sensitive to the starting point used to calculate the gradient. This phenomena well appreciated in the seismic imaging community where much work is dedicated to designing good starting points.

We study the effect of this starting model on the result of our method by adding a constant shift to the constant velocity inside the skull of the starting model. Unsurprisingly, our method degrades in quality as the starting point degrades Figure 9. This behaviour is

expected and unavoidable since our method is gradient based and nonlinear. As evidenced by failure of FWI Figure 9, this is a limitation to methods that use gradients.

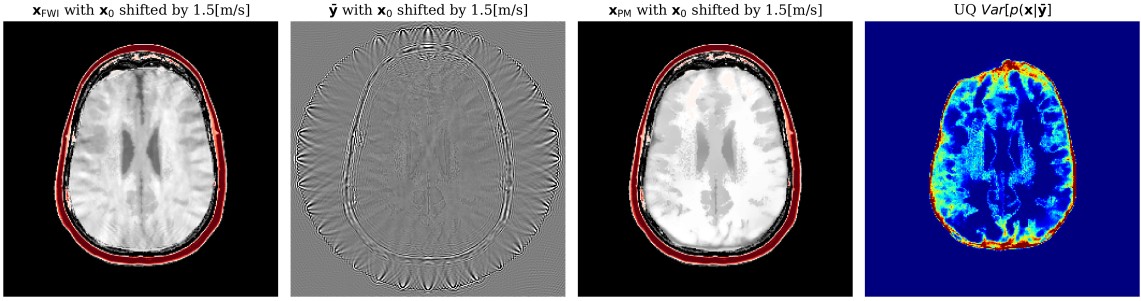

Figure 8: Evaluating our method and FWI on a starting point $\mathbf{x_0}$ that is shifted by $1.5[m/s]$ both methods still perform well

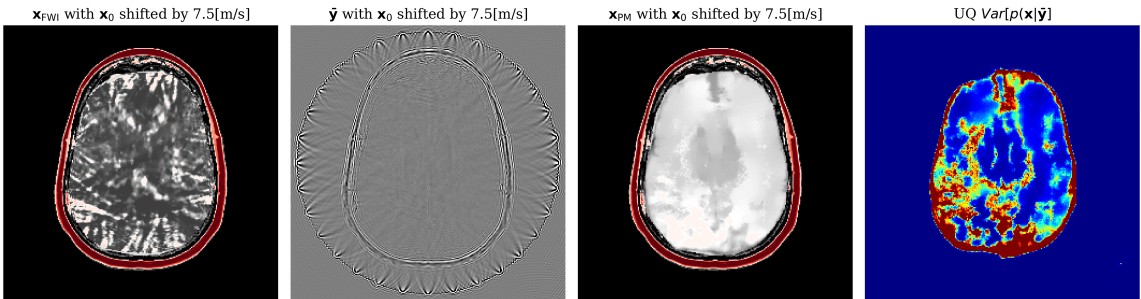

Figure 9: Evaluating our method and FWI on a starting point $\mathbf{x_0}$ that is shifted by $1.5[m/s]$. Due to the poor starting point, both methods fail. The summary statistic $\bar{\mathbf{y}}$ loses information useful for inference so our method fails as well.

### 4.6. Evaluating uncertainty

The large size of our problem, its nonlinearity and the non-Gaussianity of our prior prevents us from a comparing against a ground truth posterior. Instead, we follow the literature and evaluate our method using metrics designed the analyse the validity of the posterior from a Bayesian sense and a practical sense.

We use the following two metrics to evaluate the quality of our uncertainty:

**(i) Calibration**: On expectation, the error made by our method should correlate with the uncertainty (Guo et al., 2017). We use the method from Laves et al. (2020) to visualize calibration in Figure 10(c).

**(ii) Bayesian contraction**: a Bayesian method needs to show contraction on the ground truth as more data is observed (Ghosal and Van der Vaart, 2017). Here contraction means

that more data should decrease the uncertainty. Not only that, the error made should also decrease. Qualitatively we confirm this behaviour in Figure 5 where increasing the number of source experiments decreases the overall uncertainty. Figures 10(a) and 10(b) show that over the test set, increasing sources shows Bayesian contraction. As a scalar measure of uncertainty, we use the sum of variance for all parameters.

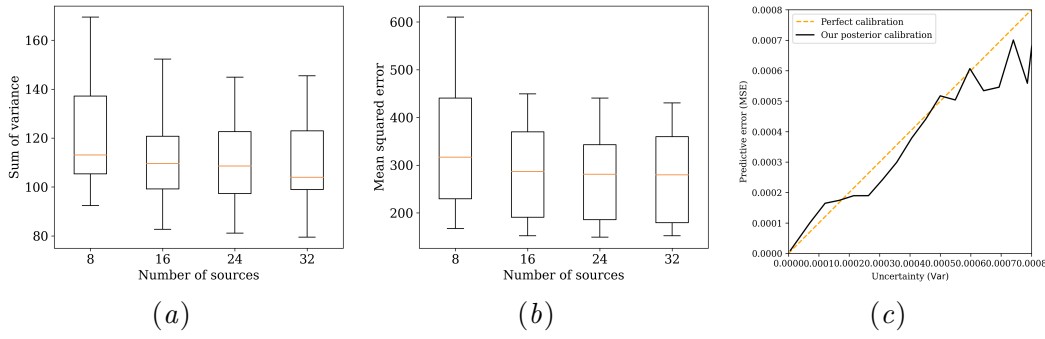

Figure 10: Validation of uncertainty quantification (a) Posterior contracts as data is increased; (b) Posterior contracts towards ground truth as measured by MSE; (c) Uncertainty correlates with error in calibration plot.

## 4.7. Selecting number of posterior samples

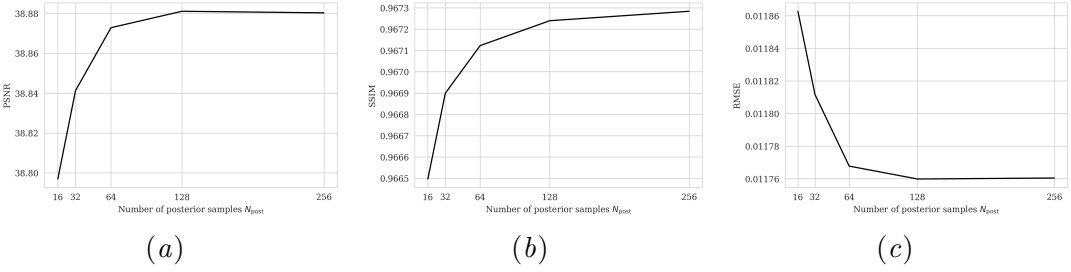

Figure 11: Effect of number of posterior samples used to estimate posterior mean on: (a) SSIM; (b) PSNR; (c) RMSE;

