# OpenReview forum: "Amortized Normalizing Flows for Transcranial Ultrasound with Uncertainty Quantification"
_MIDL.io/2023/Conference — MIDL 2023 Oral_

### Official Review · Reviewer_cBRZ · 2023-02-02

**Confidence:** 3
**Preliminary Rating:** 3

**Summary:**

The paper presents a new approach to transcranial ultrasound computed tomography.  The approach is Bayesian in nature and is based on a "normalizing flows" formulation.  The paper also uses the score function as a sufficient statistic to reduce memory usage and get all computations to fit on a GPU..

**Strengths:**

The normalizing flows idea is quite interesting, even if it is based on previous work.  It would be even more interesting if this approach were explained with more detail.

The sufficient statistics contribution is also something interesting that I hadn't seen before, even if this approach is also based on previous work.

**Weaknesses:**

Organization issues:  Figure 2 isn't numbered, and the figure references in the text don't always seem to correspond to the numbered figures.  Figures seem to be referenced out of order (for example, figure 6 is mentioned in the text before figures 4 and 5).

The reason for using Equation 2 isn't explained clearly enough.  There appears to be some kind of approximation occuring, but I am not able to understand the nature of that approximation without reading the references.  The paper fails to be self-explanatory.  This is a major flaw, since the use of Equation (2) is one of the main contributions of the paper, and I cannot appreciate the contribution if the details of the equation aren't explained.

The UQ is interesting, but seems dependent on having converged to a useful prior.  Is there a way to quantify uncertainty in the prior?  What happens to the UQ map as the size of the training data varies?

From a practical point of view, I'd much prefer the FWI image in figure 4 than the PM image.  The PM image has major hallucinations that are not present in the FWI image.  Doctors can easily look past obvious artifacts, but it's much harder to look past a realistic-looking hallucination.  Hallucination can have serious clinical consequences.

I disagree with the claim that the UQ image in Figure 4 correlates well with the error image.  The correlation actually seems quite poor, and might fail to capture the major putamen-esque hallucination that exists in the left side of the PM image.

**Deanonymize Review:**

no

**Detailed Comments:**

See previous comments

**Paper Type:**

methodological development

**Questions To Address In The Rebuttal:**

Address the weaknesses listed above.  ........................................................................................................................................................................

---

### Official Review · Reviewer_j83x · 2023-02-03

**Confidence:** 2
**Preliminary Rating:** 4
**Recommendation:** Oral

**Summary:**

The authors propose an alternative approach to transcranial ultrasound reconstruction combining neural networks with underlying physics models. Indeed, they use a neural network to model the physics of ultraosund imaging in the context of this problem. To this end, acoustic impedance fields are learned from the FastMRI dataset. The paper is well written and relevant to the conference.

**Strengths:**

* A novel framework demonstrating the power of combining deep neural networks and physics models to carry out fast, efficient and accurate  transcranial ultrasound reconstruction

* A well written paper, clear and easy to read

* Evaluation against baseline showing favourable performance

**Weaknesses:**

* How the summary statistics may need to be adjusted for changes in target population is unclear

* What is the impact of using different acquisition parameters, to capture variation between manufacturers and other sources of variation?



**Deanonymize Review:**

yes

**Paper Type:**

methodological development

**Questions To Address In The Rebuttal:**

I would suggest to clarify or expand on how the training set and the summary statistics can generalise to a larger target population; also please consider addressing the points raised in the weakness section.

---

### Official Review · Reviewer_tVpz · 2023-02-05

**Confidence:** 1
**Preliminary Rating:** 4

**Summary:**

The authors present a method using normalizing flows applied to transcranial ultrasound computed tomography to improved the speed while also quantifying uncertainly.
The make use of US physics in combination with image reconstruction techniques typically applied to other fields and evaluate this in in-silico experiments.


**Strengths:**

I am not at all an expert in US - but this paper was nevertheless quite well readable and interesting ('which I think is a strength here: )), esp the schemata are super helpful. Cant comment here in more detail I am afraid!



**Weaknesses:**

I am really not an US expert expert and not feeling able to adequately comment here, sorry!
----------------------------------------------------------------------------------------------------------------------

**Deanonymize Review:**

no

**Paper Type:**

methodological development

**Questions To Address In The Rebuttal:**

----------------------------------------------------------------------
--------------------------------------------------------------------
---------------------------------------------------------------------

---

### Meta-Review · Area_Chair_5Y47 · 2023-02-24

**Recommendation:** Accept (Oral)
**Confidence:** 4

**Metareview:**

This work proposes a novel method for transcranial ultrasound reconstruction by combining neural networks with physics models. They use a neural network to model the acoustic impedance fields learned from the FastMRI dataset. The approach is Bayesian and uses normalizing flows and score function to reduce memory usage and enable GPU computations.

It shows favorable performance compared to the baseline, and introduces interesting ideas such as normalizing flows and sufficient statistics.

The reviewers pointed out various weaknesses to the work, including unclear explanations of changes in the target population and the impact of different acquisition parameters. Also, the lack of clarity in the use of equation 2. Finally, the quantification of uncertainty (UQ) which may depend on the convergence to a useful a priori and the correlation between the UQ image and the error image which is not convincing.

All these points having been thoroughly addressed by the authors and having been integrated in the manuscript for a better understanding of the authors' work, I recommend to accept this work for MIDL.